# SIFSpec: Measuring Solar-Induced Chlorophyll Fluorescence Observations for Remote Sensing of Photosynthesis

**DOI:** 10.3390/s19133009

**Published:** 2019-07-08

**Authors:** Shanshan Du, Liangyun Liu, Xinjie Liu, Jian Guo, Jiaochan Hu, Shaoqiang Wang, Yongguang Zhang

**Affiliations:** 1Key Laboratory of Remote Sensing Science, Institute of Remote Sensing and Digital Earth, Chinese Academy of Sciences, Beijing 100094, China; 2University of Chinese Academy of Sciences, Beijing 100049, China; 3College of Geomatics, Xi’an University of Science and Technology, Xi’an 710054, China; 4Institute of Geographic Sciences and Natural Resources Research, Chinese Academy of Sciences, Beijing 100101, China; 5International Institute for Earth System Sciences, Nanjing University, Nanjing 21002, China

**Keywords:** solar-induced chlorophyll fluorescence (SIF), gross primary production (GPP), automated, in situ, measurement system

## Abstract

Solar-induced chlorophyll fluorescence (SIF) is regarded as a proxy for photosynthesis in terrestrial vegetation. Tower-based long-term observations of SIF are very important for gaining further insight into the ecosystem-specific seasonal dynamics of photosynthetic activity, including gross primary production (GPP). Here, we present the design and operation of the tower-based automated SIF measurement (SIFSpec) system. This system was developed with the aim of obtaining synchronous SIF observations and flux measurements across different terrestrial ecosystems, as well as to validate the increasing number of satellite SIF products using in situ measurements. Details of the system components, instrument installation, calibration, data collection, and processing are introduced. Atmospheric correction is also included in the data processing chain, which is important, but usually ignored for tower-based SIF measurements. Continuous measurements made across two growing cycles over maize at a Daman (DM) flux site (in Gansu province, China) demonstrate the reliable performance of SIF as an indicator for tracking the diurnal variations in photosynthetically active radiation (PAR) and seasonal variations in GPP. For the O_2_–A band in particular, a high correlation coefficient value of 0.81 is found between the SIF and seasonal variations of GPP. It is thus concluded that, in coordination with continuous eddy covariance (EC) flux measurements, automated and continuous SIF observations can provide a reliable approach for understanding the photosynthetic activity of the terrestrial ecosystem, and are also able to bridge the link between ground-based optical measurements and airborne or satellite remote sensing data.

## 1. Introduction

Accurate estimation of the CO_2_ flux uptake by terrestrial vegetation photosynthesis, that is, gross primary production (GPP), is crucial in quantifying the gas exchanges between the biosphere and atmosphere [1]. Great developments have been achieved in the field of solar-induced chlorophyll fluorescence (SIF) research in recent years. The development of high-resolution sensors deployed on aircraft [2,3] and satellites [4,5,6,7,8,9,10] especially has opened the door to detecting SIF signals at regional and global scales. SIF has been demonstrated to be an excellent proxy for plant photosynthetic function, and a high correlation between SIF and GPP has been found using remote sensing SIF products, including GOSAT SIF [4,11,12], GOME-2 SIF [13,14,15,16], OCO-2 SIF [17,18], and airborne SIF data [2,3]. 

However, the relationship between SIF and GPP is still not clear, and the effects of the temporal and spatial scale, vegetation canopy structure, and climate variability on the physiological link between SIF and GPP remain to be investigated. Therefore, in situ observations need to be used to explore this link. Techniques based on flux measurement systems have been widely used to provide CO_2_ and GPP flux data at the ecosystem level [19]. In addition, several ground-based flux observation networks have been established to combine the ground-based instruments and to define the calibration protocols and data processing procedure, which is crucial to the comparison of the flux measurements across different sites. FLUXNET is a network including hundreds of flux sites distributed across the world and is used to measure the exchange of CO_2_ and water between the biosphere and atmosphere [19]. The Spectral Network (SpecNet) was initiated with the aim to establish a global network for collecting canopy spectra coordinated with flux measurements and standardizing optical measurements across flux stations [20,21,22]. In this context, ESSEM Cooperation in Science and Technology (COST) Action EUROSPEC (ES0903) was initiated in 2009 to coordinate optical measurements on heterogeneous terrestrial ecosystems and with the aim of promoting collaboration between scientists across Europe [23]. Subsequently, ESSEM COST action OPTIMISE (ES1309) was launched based on previous European and international initiatives. Recent advances in optical sensors provide unprecedented opportunities for SIF ground-based spectral observation networks. In particular, SIF remote sensing has developed rapidly in recent decades owing to its important role in monitoring vegetation photosynthesis. 

However, long-term SIF in situ measurements are still at an early stage. Several ground-based SIF automated measurement systems have been developed along with the growing interest in in situ SIF observations in recent years. The hyperspectral irradiometer (HSI) is the first instrument dedicated to SIF detection; this instrument is equipped with two spectrometers with spectral resolutions of 1 nm and 0.1 nm. FluoSpec, which uses HR2000+ spectrometer with a spectral resolution of 0.13 nm, is one of the first instruments to obtain continuous measurements of SIF over deciduous forests [24]; however, it does not include a temperature control and the signal-to-noise ratio (SNR) is relatively low (about 250). Two similar systems, MRI and SFLUOR box, are based on the use of an optical switch, a fiber optic multiplexer (MPM-2000-2 × 8-VIS) [25]. Two spectrometers, HR4000 with a spectral range of 400–1000 nm and a spectral resolution of 1 nm, and SPEC_Fluo_ with a spectral range from 700 to 800 nm and a spectral resolution of 0.1 nm, are embedded in these systems [25]. FluoSpec 2 replaces the HR2000+ spectrometer with a QEpro spectrometer, which has a higher SNR and a better temperature control and data collection flow compared with the FluoSpec instrument [26]. However, the spectral domain of FluoSpec 2 ranging from 730 nm to 780 nm covers only the near-infrared region. In recent years, an SIF network (FluoNet) consisting of several automated SIF systems based on the use of FluoSpec and FluoSpec 2 instruments has been established in the USA. FAME was recently designed using a specific QEpro with a spectral resolution of 730–786 nm and a spectral resolution of 0.15–0.17 nm [27]. One distinct innovation of FAME is the use of a datalogger as a control unit, allowing SIF and supporting environmental measurements to be precisely synchronized and easily integrated with existing data acquisition systems at eddy covariance (EC) tower sites [27]. However, all of the existing instruments mentioned above ignore the important functions of SIF at the red band. Another system, the fluorescence box (FLox) is installed in a portable tower and collects reflectance and SIF measurements at Optimizing Production Inputs for Economic and Environmental Enhancement (OPE3) site, which is covered by corn [28]. The FLox system consists of two spectrometers, including an Ocean Optics FLAME S (400–900 nm) and an Ocean Optics QEpro (650–800 nm), which has a FWHM of 0.3 nm. However, the effects of the atmosphere on SIF retrieval using tower-based measurements, especially for forest observations, are assumed to be negligible by currently used systems [29]. It is worth mentioning that the recent work has also addressed the importance of oxygen compensation on proximal sensing SIF retrieved at the O_2_ band using Fraunhofer line discrimination (FLD) and spectral fitting methods (SFM) [29]. This method was used in the field measurements for a boreal forest ecosystem at a height of about 18 m to account for the energy reabsorption effects on SIF retrievals at the O_2_–A band [30].

In order to provide a reliable approach for collecting long-term SIF measurements in combination with continuous EC observations based on China’s CO_2_ flux observation network, ChinaSpec [31], an SIF observation network, was established in 2017 by several research groups, so far mainly from Chinese Academy of Sciences and Nanjing University. This network covers a variety of terrestrial ecosystems, including different types of agricultural, grassland, and forest ecosystems distributed across the mainland of China. 

This paper presents the details of the system components, instrument installation and calibration, as well as data collection and processing for the automated in situ SIF measurement (SIFSpec) system produced by Bergsun Inc. (Beijing, China) and used in the ChinaSpec network, along with two years of continuous SIF measurements on a maize crop at the Daman (DM) site. The objectives of this paper are as follows: (1) to present an overview of the composition and installation of the automated in situ SIF measurement system; (2) to introduce the data collection and processing procedure, especially for atmospheric correction; and (3) to test the performance of the SIFSpec system for SIF measurements and GPP estimation. 

The remaining parts of this paper are organized as follows. The details of the system components and data collection are provided in Section 2; descriptions of the site, instrument installation, and data processing are given in Section 3; long-term measurements and their links with crop GPP at the DM site are presented in Section 4; finally, the discussion and summary are presented as Section 5 and Section 6, respectively.

## 2. System Description and Data Collection

### 2.1. System Design

The SIFSpec system mainly consists of the central control unit, a spectrometer with high spectral resolution and SNR, the light collection unit, and the temperature control unit. A schematic diagram of the configuration of SIFSpec is shown in Figure 1. All units (except the optical fibers) of the system are housed in a box with dimensions of 56 cm × 35 cm × 29 cm.

The core of the central control unit, an industrial personal computer, is responsible for data logging, control, collection, processing, and storage. The computer is equipped with a random access memory (RAM) of 2 GB and a read-only memory (ROM) of 256 GB. This memory is sufficient for saving measurements made during a one-year period as the size of the daily data file is generally no more than 40 MB if the data-sampling time interval is set as 3 min. Several pieces of software are included in the system for collecting, calibrating, browsing, displaying, and transforming the data.

The spectrometer used in the SIFSpec system is a QE65 Pro spectrometer (Ocean Optics, Dunedin, FL, USA). The spectral resolution is determined by both the dispersion and pixel resolution, that is, the characteristics of both the grating and slit that are selected. The QE65 Pro used in this system is customized with a high spectral resolution of around 0.34 nm using a longpass filter (>590 nm), and an H6 grating (GRATING_#H6, Ocean Optics, Inc., Dunedin, FL, USA) with a groove density of 1200 mm^−1^ and a 25 μm slit (INTSMA-025, Ocean Optics, Inc.). The spectral sampling interval is about 0.155 nm. The spectral range of 649–805 nm and peak signal-to-noise ratio (SNR) of over 1000 make it possible to retrieve the full-band-width SIF spectrum with two peaks at 685 and 740 nm. The QE65 Pro spectrometer is equipped with a back-thinned full frame transfer Charge-Coupled Device (FFT-CCD) detector with 1044 × 64 pixels and a 16-bit A/D converter. The QE65 Pro has a 2D arrangement of pixels (1024 × 58) and a peak quantum efficiency of around 90%. The readout noise can be minimized and the processing speed can be enhanced by binning or summing the detector’s columns as a pixel prior to the readout process. Additionally, the thermal noise is also reduced by the onboard Thermal Electric (TE) cooler. Thus, a superior dynamic range of 65,000:1 for a single acquisition can be delivered thanks to the combination of the QE65 Pro’s low noise detector and 16-bit A/D converter, and the sensitivity can be up to ~0.065 counts/e-. The amount of stray light can be lower than 0.08% at 600 nm and is 0.4% at 435 nm. In order to protect the spectrometer, a black customized thermotank is designed to separate the spectrometer from other parts of the SIFSpec system.

The light collection unit is composed of a bifurcated optical fiber (CPATCH, Ocean Optics, Inc.), which has a bundle core 1000 μm in diameter and two electronic TTL shutters that are used to collect the up-welling and down-welling radiation. The bifurcated optical fiber is a Y-shaped subassembly with two parallel fibers that have the same diameter and that merge to one in the middle. One 5 m long optical fiber is fixed in the upward direction and another is fixed in the downward direction. In the upward direction, the downward irradiance is measured using a cosine-corrected foreoptic (CC3-UV-S, Ocean Optics, Inc.), which can capture hemispherical measurements with a large field of view (FOV) of 180°. The conical foreoptic uses a bare fiber with a small FOV of about 25° and is used to capture the upward radiance when pointing at the cropland canopy. Two electronic TTL-driven shutters (INLINE-TTL, 140 mm × 50 mm × 50 mm) are employed to connect the bifurcated optical fiber and two 5 m long bare fibers, which means that light path can be blocked without disturbing the measurement process. 

The temperature control unit is designed to maintain an ideal temperature using a thermoelectric cooler, a temperature transducer, and a Proportion Integration Differentiation (PID) controller. The SIFSpec system is designed to be mounted at different sites covering a variety of terrestrial ecosystems, meaning that the climate may differ greatly at different sites. In order to guarantee that the spectrometer’s CCD can operate normally, as well as to control a main source of the thermal noise, the temperature inside the system has to be controlled. The thermoelectric cooling system consists of a TE cooler; two heatsinks; and two fans, one placed outside and one placed inside the thermotank. The TE cooler works to keep the temperature inside the thermotank at 25 ± 1°. In addition, the TE cooler inside the spectrometer can cool the CCD detector down to −15 °C. 

### 2.2. Data Collection

The central controlling software includes several modules that implement different functions. An open−close module is required to manage the computer and the measurement process. The solar zenith angle (SZA) is calculated based on the latitude, longitude, and the local time so that the time when the measurements should start and stop can be determined. Once the SZA is larger than 85°, the spectra acquisition process will stop. 

Two alternative observation modes, manual and automatic, are provided. The automatic mode is used mainly during the making of continuous, long-term field measurements. The manual mode is more suitable for use during the calibration process or when making a single measurement for a specific purpose. In the automatic observation mode, the single beam sandwiched method is employed to collect the down-welling and up-welling spectra following the method proposed by the previous works [25,32]. The schematic observation sequence is displayed in Figure 2. The first down-welling solar irradiance, the up-welling radiance, and the second down-welling irradiance are sequentially collected by controlling the shutters. The effect of the variation in the solar irradiance resulting from the time delay can be reduced by averaging the two down-welling irradiance measurements. Typically, an entire measurement can be accomplished within 15 s at noon and within 2 min at sunrise and sunset.

The integration time is a key parameter for determining the amount of light entering the spectrometer via the optical fibers and has to be set corresponding to the incident irradiance. In order to maximize the SNR, the incoming energy has to make full use of the dynamic range of the spectrometer. A range for the ratio of the measured intensity to the saturation value of the spectrometer is set to automatically determine the integration time. A simple integration time optimization method ([33]) is used in this system. Iterations are not required for this algorithm and this helps to reduce the time delay to as little as possible.
(1)$${\mathrm{IT}}_{opt}=\frac{DN_{opt}}{DN_{default}}IT_{default},$$
where ${\mathrm{IT}}_{opt}$ is the optimized integration time and $IT_{default}$ is the default integration time measured previously. $DN_{default}$ is the maximum value of the spectral range at the default integration time. $DN_{opt}$ is the optimized count value with a variable range from 80% to 90% of the spectrometer saturation value (~65,000). In general, the integration times are around 600 ms and 500 ms at noon under clear-sky conditions for the downward and upward channels, respectively. However, the integration time will significantly increase under low light conditions, especially in the early morning and late afternoon or if it is cloudy, which will increase the risk of a time mismatch between the downward irradiance and upward radiance measurements. The solar irradiance, in particular, fluctuates rapidly as a result of moving clouds, as well as in the early morning and late afternoon. This can cause a large bias in the retrieved spectral reflectance and, as a result, there is the potential for making inaccurate SIF retrievals. Thus, a maximum integration time (IT) of 8 s is set for any illumination conditions. By default, the IT is about 0.6 s at noon and 3 s in the morning and afternoon for the down-welling channel on a cloud-free day. For the up-welling channel, the IT is around 0.4 s and 1.8 s at these times.

The dark currents (DCs) resulting from the thermal noises and readout noises continuously vary with the integration time and the temperature. Thus, the DCs are recorded for each measurement for post-processing if needed. The time delay caused by the within-sequence DC correction is negligible.

## 3. Field Measurements and Data Processing

### 3.1. Field Installation at EC Flux Sites

Up to now, the SIFSpec systems have been in operation at 10 EC flux sites, including five cropland sites, two forest sites, and three grassland sites. Details of the sites, including the location, height of instrument above the ground, and observed ecosystem type, are listed in Table 1.

The SIFSpec system is usually mounted on the eddy covariance (EC) tower. A 4 m long square iron welded pipe, which extends 3 m away from the boundary of the platform, is used to fix the two optical fibers and also to ensure that the cosine-corrector on the upward optical fiber points vertically upward. The pipe is fixed on the south side using several detachable U-shaped extending pieces of irons when loading and unloading the instruments. A specially designed stainless board is mounted at the end of the pipe to fix the angle of the optical fibers. This board has an aluminum alloy spirit level on the top to monitor the angle of the board and an angulometer welded on the upper right side to control the orientation of the down-welling fiber. The bare fiber points at the vegetation canopy with a view zenith angle of about 25° in order to avoid the heterogeneous or sparsely vegetated surface below the EC tower.

As for sites where there is no EC tower, the design concept for the instrument installation is different. A customized double-layer shelf is employed to place the instrument. The height of the shelf is about 2 m and the SIFSpec is placed on the upper layer to avoid the influence of the ground moisture and crop irrigation. A manual bracket with cantilevers is used to mount the optical fibers. Both the telescopic pole and the telescopic arm have three sections, which are used to regulate the height and the length. The telescopic arm is normally set to extend 3 m from the telescopic pole. The height of the telescopic pole set up is 2.5 m for wheat or grassland measurements and 4 m for maize observations.

Meteorological and radiation sensors for long-term observations are deployed on the flux tower at each site in the ChinaSpec network. Meteorological data are obtained every 10 min and processed into quality-controlled 30 min and daily averages [34]. The surface soil heat fluxes; photosynthetically active radiation (PAR); broadband incoming shortwave radiation ($R_{in}$); thermal radiation of the cropland; and meteorological parameters including air temperature (Ta), air pressure (p), vapor pressure deficit (VPD), air humidity (Rh), wind speed (u), four-component radiation, and turbulent fluxes are simultaneously measured using the specific instruments deployed on the tower at different heights [35].

The EC flux data, including the exchange of CO_2_, energy, and H_2_O, are collected by the EC system with a sampling frequency of 10 Hz [36]. The raw EC data are processed to obtain the atmospheric CO_2_ concentration (Ca), net ecosystem exchange of CO_2_ (NEE), sensible heat flux (H), latent heat flux (LE), and friction velocity (u*) using the Edire software [36,37]. All data are converted to 30 min and daily values corresponding to the SIF time steps. The meteorological and EC data collected on days with corresponding SIFSpec measurements were selected for future study.

### 3.2. Spectral and Radiometric Calibrations

The complex environment experienced during the long-term unattended in situ measurements will degrade the performance of the spectrometer, optical fibers, and cosine-corrected foreoptics. Thus, regular calibration needs to be carried out to maintain the quality of the observed spectra, especially for the SIF retrieval. This is particularly relevant to the precise location of the narrow absorption band and to the absolute irradiance and radiance. Spectral calibration is regularly carried out by the SIFSpec manufacturer using the method described by the spectrometer manufacturer (Ocean Optis, Inc., Dunedin, FL, USA).

Absolute radiometric calibration is regularly conducted to maintain the performance of the instruments. In general, the calibration is conducted before the instrument installed for every growth period for cropland site and once a year for the forest site. To obtain the precise radiometric coefficients, the laboratory radiometric calibration method is used for this system. As with traditional laboratory calibration, an integrating sphere, Labsphere XTH2000C (Labsphere Inc., North Sutton, NH, USA), is used as the standard light at the calibration laboratory of the Institute of Remote Sensing and Digital Earth, Chinese Academy of Sciences. The 20 inch diameter integrating sphere has an 8 inch diameter port, which is coated with Spectraflect^R^ white coating, giving near Lambertian properties and resulting in a better spectral reflectance of 0.99 in the range from 400 nm to 1100 nm and a better radiance uniformity of 98%. The broadband unpolarized light covering the spectral range from 350 nm to 2400 nm is generated with a wide radiant brightness range, which varies from 0 to 7000 cd/m^2^. The radiant temperature ranges from 3299 to 6000 K.

The optical fiber containing the cosine-corrected foreoptic and bare fiber are simultaneously placed at the entrance aperture of the integrating sphere, as shown in Figure 3. Once the IT is optimized by confirming that the maximum value measured across all of the bands exceeds 80% of the saturated value, five measurements are made in sequence using the manual observation mode. Next, the DCs for the corresponding IT are recorded by opening the shutter of the light channel. After that, the calibration coefficients K(λ) for the upward and downward channels are calculated using Equation (2):(2)$$K\left(\lambda \right)=\frac{L_{light}\left(\lambda \right)\cdot IT}{\left(DN\left(\lambda \right)-DC\left(\lambda \right)\right)},$$
where DN(λ), DC(λ), and IT are the raw digital count values, dark current, and integration time, respectively, corresponding to each measurement. $L_{light}$ is the radiance of the calibration light source.

The results of the calibration of one system at the DM site, conducted in the laboratory in two different years, are displayed in Figure 4. It can be observed that the calibration factors are almost the same for these two years, which indicates the overall stability of the instrument. However, the calibration factor for the downward channel in 2019 is slightly different from that in 2018. This can be partly explained by the degradation of the cosine-corrected foreoptic owing to the long-term exposure to the field environment. It also indicates the necessity for annual radiometric calibration and maintenance.

### 3.3. Data Processing

The data processing chain for the radiance and irradiance measurements collected by the SIFSpec system includes quality filtering, atmospheric correction, and reflectance and the retrieval of the SIF. The down-welling irradiance and up-welling radiance spectra are collected approximately every three minutes, which results in a large volume of spectral measurements. Thus, automated processing is essential for efficient retrieval of the SIF results.

Firstly, quality filtering of the raw data is carried out to provide a high-quality SIF dataset. Two main criteria—the stability of measurements between two irradiance observations and obvious SIF contributions to the apparent reflectance—are included in the quality control. The mismatch between the downward irradiance and upward radiance observation times is the primary problem that needs to be considered. The data will be rejected for use in SIF retrieval if the SZA is larger than 85°, as a larger SZA will increase the risk of there being a larger variation in the down-welling irradiance. In addition, the difference between the first and second downward irradiance values found during one whole measurement is used as a quality indicator, with a threshold value of 10% being used to directly reject data where there is a time mismatch between the measurements made by the two channels. The peak in the apparent reflectance is useful for further data quality control. Data with an obvious peak in the apparent reflectance will be retained for subsequent SIF retrieval. Furthermore, the spectral shift problem between upward and downward channels can also be identified using the ‘dip’ peak in the apparent reflectance. The spectral shift will obviously influence SIF retrievals that are made using FLD-based methods [38].

Secondly, an automatic software module is set up to conduct atmospheric correction on the long-term SIF system observations. The O_2_–A and O_2_–B bands are commonly used to retrieve SIF results based on long-term tower-based measurements. In order to obtain accurate SIF retrievals at the top of the canopy, the up-welling radiance and down-welling irradiance collected at the flux tower should be corrected to those at the top of canopy, as shown in Figure 5. According to the analysis made by previous studies [39,40], for tower-based measurements, if the height of the flux tower is greater than 10 m, an SIF retrieval bias of about 0.1 mW m^−2^ sr^−1^ nm^−1^ at the O_2_–A band will be introduced as a result of the radiative absorption and scattering between the sensor and the canopy. Compared with the absolute intensity of the SIF signal (<2 mW m^−2^ sr^−1^ nm^−1^), this bias is not negligible. Thus, it is important to carry out atmospheric correction on both the tower-based down-welling irradiance and up-welling radiance before the SIF retrieval process, especially for the O_2_–A band [41], which has a lower transmittance.

According to the method proposed in the previous work [39], a simple and operational method based on look-up-tables (LUTs) can be used to estimate the atmosphere transmittance between the tower-based instrument and the top of the vegetation canopy for both the downward irradiance and upward radiance. Atmospheric transmittances at the O_2_–A and O_2_–B bands are mainly related to the aerosol optical depth (AOD), radiative transfer path length (RTPL), atmospheric pressure, and temperature [42]. To establish the look-up-tables for a specific site, the parameter dataset, which includes the height of the instrument, altitude, view zenith angle (VZA), and a range of solar zenith angles (SZA) and values of the AOD, is needed in simulations made using the MODerate resolution atmospheric TRANsmission 5 (MODTRAN 5) model. For in situ observations, AOD_550_ is calculated using the ratio of the down-welling irradiance at 790 nm to that at 660 nm. In this study, the upward vegetation radiance was measured using a bare optical fiber; this radiance is different from the downward irradiance measured with the cosine-corrector. Thus, the look-up tables for the upward and downward channels were constructed separately; and the tower-based SIF retrievals at O_2_–A band before/after atmospheric correction are presented in Figure 6. It shows that the SIF results are obviously higher after atmospheric correction than the uncorrected SIF results, especially for noon time. For details, please refer to Liu et al. (2019) [39].

Thirdly, several ground-based SIF retrieval algorithms based on the FLD principle are widely used. Four different single-band SIF retrieval results at the Hα (656 nm), O_2_–A (761 nm), and O_2_–B bands (688 nm) based on standard FLD (sFLD) [43], 3-band FLD (3FLD) [44], improved FLD (iFLD) [45], and principal component analysis-based FLD (pFLD) [46] algorithms are simultaneously obtained for the processing of SIFSpec measurements. In addition, the full-spectrum spectral fitting method (F-SFM) [47] based on principal components analysis (PCA) is employed to acquire the full-spectrum SIF retrievals. An example of single-band diurnal SIF curves at the O_2_–A band (measured on 7 September 2017) is illustrated in Figure 7. This example shows that there is no obvious difference between the results obtained using the different SIF retrieval methods. The 3FLD method is considered to be relatively simple and robust for ground-based measurements with a spectral resolution of 0.3 nm [40,48]. Therefore, the 3FLD algorithm was selected for the SIF retrieval in this study. Moreover, common VIs, including the Normalized Difference Vegetation Index (NDVI), the MERIS Terrestrial Chlorophyll Index (MTCI), and the NIR reflectance of vegetation (NIRv), were also simultaneously calculated using the measured reflectance data.

The online tool available on the Max Planck Institute for Biogeochemistry (MPI-BGC) website (http://www.bgc-jena.mpg.de/~MDIwork/eddyproc/) can be used to calculate the GPP data using the input parameters Rin, Rh, LE, H, Ta, and u* based on the night time-based partitioning algorithm [49,50]. GPP is calculated by separating the day-time NEE and the estimated respiration. The gap filling uses the light response function (Michaelis–Menten equation) and respiration function (Van’t Hoff equation) [49,51]. GPP data with 30-minute time intervals are calculated for analysis in conjunction with the SIFSpec measurements. The daily average of GPP is also calculated using the 30-minute data.

## 4. Data Analysis

In order to investigate the reliability of the measurements acquired by the SIFSpec system, the diurnal and long-term observations at the DM site are selected as an example for analysis in this paper.

### 4.1. Diurnal Variation in Radiant Spectra and Apparent Reflectance

Several down-welling and up-welling radiance spectra covering the period from morning to afternoon observed on a clear day (7 September 2017) at the DM site are shown in Figure 8. The spectral range is from 649.3 nm to 805.4 nm, thus covering the full SIF emission spectrum. It can be seen that the absorption depth of the incoming radiation has distinct diurnal variations, which is consistent with our prior knowledge that the absorption depth of the radiation spectrum is greater in the morning and afternoon than that at noon, which can be explained by the increased radiation transfer path length for larger SZAs [52]. Also, the fraction of diffuse radiation is higher in the early morning and late afternoon than at noon time, and the transfer path of the direct radiation is shorter than that of the diffuse radiation, which results in a deeper absorption line for the diffuse radiation than for the direct radiation [53].

The diurnal variations in the apparent reflectance spectrum (the ratio of up-welling to down-welling radiance) are also depicted in Figure 8. The approximate PAR values corresponding to the spectra are 648, 1283, 1525, 1277, and 947 μmol m^−2^ s^−1^. A peak due to the in-filling effect of the SIF signal can be observed in the apparent reflectance spectra in the range from approximately 760 to 762 nm. The peak in the apparent reflectance contributed by the SIF is also an indicator for the data quality filtering—an obvious peak in the apparent reflectance indicates that the measurements are reliable. Comparing the diurnal variations in the apparent reflectance data, it can be seen that the magnitude of the apparent reflectance is larger in the morning and afternoon and smaller at noon. This is related to the bidirectional reflectance distribution function (BRDF) characteristics of the vegetation canopy, which result in different diurnal variations in reflectance for the direct and diffuse radiation [39,54,55].

Furthermore, the relative height of the peak in the O_2_–A band is smaller at noon than that in the morning and afternoon, which is counterintuitive because the higher SIF value should produce a stronger in-filling effect for the same absorption depth. However, in the morning and afternoon, a stronger relative contribution of in-filling of the SIF to the apparent reflectance can be observed from the apparent reflectance than at noon. Three reasons or hypotheses can be used to explain this phenomenon. Firstly, although the absolute SIF value is larger at noon than in morning or afternoon, a smaller SIF excursion peak is present in the apparent reflectance because the depth of the absorption line is smaller at noon than in the early morning or late afternoon. Secondly, there is the negative in-filling effect of the direct radiation at noon [55]. According to the study in the previous work [55], both the SIF in-filling effect and the direct direct radiation contribute to the height of the peak in the apparent reflectance. Another possibility is the decreases in the SIF yield efficiency and the fraction of absorbed photosynthetically active radiation (fPAR) that occur at noon. Hu et al. [56] pointed out that the SIF inside the absorption line can be calculated by multiplying the height of the peak of the apparent reflectance and the incoming solar radiation inside the absorption band. Thus, the decrease in the height of the peak at around 760 nm may also be the result of the decrease in the product of the SIF yield and fPAR. In general, the value of fPAR is lower at noon than in the morning and afternoon, but the diurnal variations in SIF yield need to be investigated in future research.

### 4.2. Diurnal Response of SIF at both O_2_–A and O_2_–B Bands to PAR

Figure 9 displays the diurnal collections of SIF and PAR obtained by the SIFSpec and EC systems at the DM site. Four different days (day of year (DOY) = 155, 177, 205, and 250; 2018), corresponding to the jointing, booting, tassel, and milk stages of one growing season, have been selected to depict the diurnal variations in SIF at the DM site.

It can be seen from Figure 10 that in the overall iurnal patterns in SIF values at both the O_2_–A and O_2_–B bands and in the incoming PAR, higher values around midday and lower values in the early morning and late afternoon are consistent. The diurnal pattern for SIF curve is also consistent with that for the PAR when the maize canopy is at the fully developed stage, especially the O_2_–A band. It can be observed that at both the O_2_–A and O_2_–B bands, the SIF becomes almost saturated when the incident PAR is high at midday during the growing stage.

The differences in the diurnal response of SIF to PAR at different growing stages are shown in Figure 10, which displays the diurnal variations in this response at the DM site on the same four days as in Figure 9. The overall consistency in the diurnal SIF values at both the O_2_–A and O_2_–B bands and in the incoming PAR is clear. Again, the PAR values are higher around midday and lower in the early morning and late afternoon.

Further observation of the diurnal variations in SIF show that, for the same PAR level, the SIF value at the O_2_–A band value is lower in the afternoon than in the morning during the growing season. In the case of a fully developed canopy, that is, at the tassel and milk stages (DOY = 205 and DOY = 250; 2018), there is no obvious difference in the response of SIF to PAR. However, this phenomenon will be sustained across the whole growing season for the O_2_–B band. Three hypotheses that attempted to explain this phenomenon were proposed in the previous study [27]. These included the stronger nonphotochemical quenching (NPQ), reduced light interception due to the repositioning of chloroplasts, and the changes in the distribution of the leaf orientation distribution in the afternoon as compared with the morning. However, the different responses of SIF to PAR at the O_2_–A and O_2_–B bands when the canopy is at the fully developed or senescent stages should be further investigated in future studies.

### 4.3. Seasonal Variations in SIF, NDVI, and GPP

In order to monitor the maize during two consecutive growing cycles, the SIFSpec system was first installed on the tower at the DM site on 13 May 2017 when the maize had just sowed. However, the instrument only began operating normally on 6 June 2017, and two obvious measurement problems occurred—one from 12 to 19 July 2017 and one from 9 to 21 August 2017. These were the result of the breakdown of the SIFSpec system hardware. In 2018, the instrument was mounted prior to the corn sowing in 2018 and no obvious measurement problems occurred during this growing cycle. Therefore, almost continuous measurements were collected at the DM site during these two growing cycles, especially in 2018. A total of 208 days of observations was obtained at the site during 2017 and 2018.

The SIFSpec measurements were acquired roughly every 3−5 min from 07:00 to 19:00 local time. This time interval is made up of the predefined three minutes as well as the time needed to complete a whole measurement cycle, which includes the collection of one irradiance spectrum and two radiance spectra. The IT optimal time before each sampling and the time for shutter switches make up the rest of the 3−5 min. Thus, more than 250 spectra were collected on clear-sky days. The SIF retrievals were averaged into half-hour values after quality control. In order to explicitly present the seasonal variability in SIF in tandem with PAR, NDVI, and GPP, all variables were processed into daily averages. The long-term, continuous measurements made by the SIFSpec system, as well as flux data covering two almost complete growing seasons at the DM site, are depicted in Figure 11.

The NDVI is a good indicator of the vegetation growth. The variation in SIF at the O_2_–A and O_2_–B bands with NDVI are presented in Figure 11a,b, respectively. Overall, the seasonal dynamics of SIF values at both the O_2_–A and the O_2_–B bands are consistent with the variation in NDVI. A better correlation between the SIF and NDVI can be observed at the O_2_–A band than that at the O_2_–B band; the corresponding correlation coefficients are 0.58 and 0.37, respectively. The SIF values tend to show distinct rising trends earlier than NDVI at the growing stage of the 2018 season.

A pleasing synergistic variation in the GPP data and SIF at both bands can be observed. Compared with the SIF at the O_2_–B band, the SIF at the O_2_–A band is more significantly correlated with GPP, with a correlation coefficient of 0.81. The increases and decreases in SIF and GPP are both consistent with canopy growth and senescence during the two growing seasons. As for the O_2_–B band, a much better variation of SIF with GPP can be observed for 2018 than for 2017. The reliable correlation between SIF and GPP indicates the ability of SIF to track the photosynthetic activity of vegetation.

## 5. Discussion

### 5.1. Influence of the Use of a Cosine-Corrected Foreoptic

Hemispherical measurements enable the sampling of a wider area [23], and so a cosine-corrected foreoptic can also be used to collect the up-welling radiance reflected by the vegetation. The resulting spectral measurement footprint matches better with the eddy flux observations. Liu et al. [42] have stated that for hemispherical up-welling observations using a cosine-corrected foreoptic, 90% of the total radiation comes from a field of view (FOV) of 144°, which corresponds to 75.68% of the area of the flux measurements. Therefore, for a heterogeneous forest ecosystem, a cosine-corrected foreoptic should be used to collect the up-welling radiance. According to the statistical results [57] from 2011, about one-third of the observation systems at flux towers use hemispherical observations of the upward vegetation irradiance. The up-welling radiance is collected using a bare fiber for crop and grassland sites, whereas a cosine-corrected foreoptic is used for forest sites.

However, there are also some disadvantages of using a cosine-corrected foreoptic to collect the up-welling radiance. Firstly, the large light-loss fraction (about 70% at the far red band) increases the risk of a mismatch between the two channels because the detector integrating time is longer under weak illumination. Secondly, the SNR will slightly decrease thanks to the low transmittance of the cosine corrector, which will influence the quality of the SIF retrieval results [42]. Thirdly, the shadow of the tower will be included in the hemispherical field of view. The size of this effect will vary with the solar zenith angle and influence the potential application of the observations [23]. Finally, the calculation of the radiation transfer path is more complex for hemispherical observations, which is an important consideration when conducting atmospheric corrections. Liu et al. [42] found the equivalent radiance transfer path length to be twice the height of the sensor for tower-based hemispherical observations. However, this was based on assumptions that ignored the heterogeneity of the surface and the directional characteristics of the SIF and reflectance [42].

### 5.2. Uncertainties in SIF Retrieval Caused by Time Mismatch Between Measurements

The light path-switching approach was employed in this SIFSpec system using a single spectrometer to sequentially measure the up-welling and down-welling radiation fluxes. Making use of a single spectrometer has several advantages compared with the use of two spectrometers. It will result in a lower-cost, lighter system and there will also be no need for regular inter-calibration of the two spectrometers. However, there are also several unavoidable disadvantages. The problem of non-simultaneous up-welling and down-welling measurements is the biggest issue.

The mismatch between the time of the upward radiance and downward irradiance measurements is a major problem in long-term in situ SIF observations [27]. The widely used ground-based SIF retrieval method, 3FLD, suffers from the problem that the SIF values cannot be accurately retrieved if the weather conditions vary quickly when the upward radiance and downward irradiance are being measured. In contrast, the data-driven method [58,59] is an alternative method that can be used for long-term in situ SIF observations to avoid the need for downward irradiance measurements. This is based on the idea that irradiance spectra can be reconstructed using a linear combination of several principal components (PCs) using the SIF-free spectra. However, the SIF retrievals based on the data-driven method are also sensitive to the training dataset selected to produce the PCs.

### 5.3. Effects of Atmospheric Correction on SIF Retrieval

The SIFSpec system is generally mounted on the flux towers at a height ranging from 5 m to 50 m depending on the specific ecosystem at a given site. As Sabater et al. [29] pointed out, the 3FLD method widely used in ground-based SIF retrieval will underestimate the SIF if the atmospheric correction effects at oxygen absorption bands are not compensated for. Recent studies [40,42] also quantitatively demonstrated the effects of atmospheric correction on the SIF retrieval for tower-based observations. According to the results of field experiments conducted on tower-based measurements [39], after atmospheric correction, the root-mean-square error (RMSE) decreased from 0.221 to 0.078 mW m^−2^ sr^−1^ nm^−1^. Thus, it is necessary to use the atmospheric correction method based on an LUT used for long-term observations by the SIFSpec system for more accurate retrieval of SIF from tower-based observations.

### 5.4. The Selection of the Spectral Range for the SIFSpec System

At the present stage, although the spectral range of spectrometers used in the SIFSpec system is about from 650 nm to 800 nm, the other existing automated systems mainly cover the near-infrared (NIR) region (~730–780 nm), as introduced above. Thus, the information of red SIF retrieval is not available for other SIF observation systems only covering the NIR region. In practice, it is more difficult for red SIF signals to escape from canopy because of the strong re-absorption and scattering effects inside the vegetation canopy [60]. Therefore, the red SIF is much weaker and is also more difficult to measure at canopy level than that at the NIR band, which may be the main reason to neglect the red band in the systems. However, the red SIF may be more directly linked to photosynthetic activity because it contains more information of the photosystem II [60,61,62]. In addition, although the spectral resolution will decrease for the instrument with a wider spectral range (concluding red and NIR bands) compared with a shorter range (only concluding NIR band), which decreases from 0.15 nm to 0.34 nm, the integration time will be reduced and the SNR will increase to some extent because of the increased photons that reach each detector per unit time [26]. The previous work has also demonstrated that the difference of the retrieved SIF with spectral resolution of 0.1 nm and 0.3 nm at the O_2_–A band can be negligible if tested using the 3FLD SIF retrieval method [48]. Furthermore, the red SIF signal will be much more enhanced if it is downscaled from the canopy level to the photosystem level [63], which may improve the correlation with photosynthetic activity. Therefore, in this context, the red band is suggested to be concluded in the SIFSpec system.

## 6. Conclusions

Automated and long-term in situ SIF measurements coordinated with meteorological and EC observations have a significant role in terrestrial ecosystem research. In particular, long-term in situ SIF observations provide unprecedented opportunities for the direct validation of satellite-based and airborne-based SIF retrievals and the possibility of investigating the physiological link between SIF and GPP. In this paper, the automated and long-term in situ SIF measurement system—SIFSpec used in the ChinaSpec network—was presented. Compared to most currently existing SIF observation system only covering the NIR region, which can only retrieve an NIR SIF signal, our SIFSpec can record both red and NIR SIF signals. We have demonstrated the reliability and practicability of the SIF measurements collected by the SIFSpec system using the observations made at the DM site over two growing cycles. The consistency between the diurnal variations in SIF and PAR at both the O_2_–A and O_2_–B bands throughout the entire canopy growth stage is essential to the psychological mechanism of SIF emitted by vegetation. The consistency between the seasonal variations in SIF and GPP demonstrates the important role of SIF in the exploration of the photosynthetic function of vegetation.

## Figures and Tables

**Figure 1 sensors-19-03009-f001:**
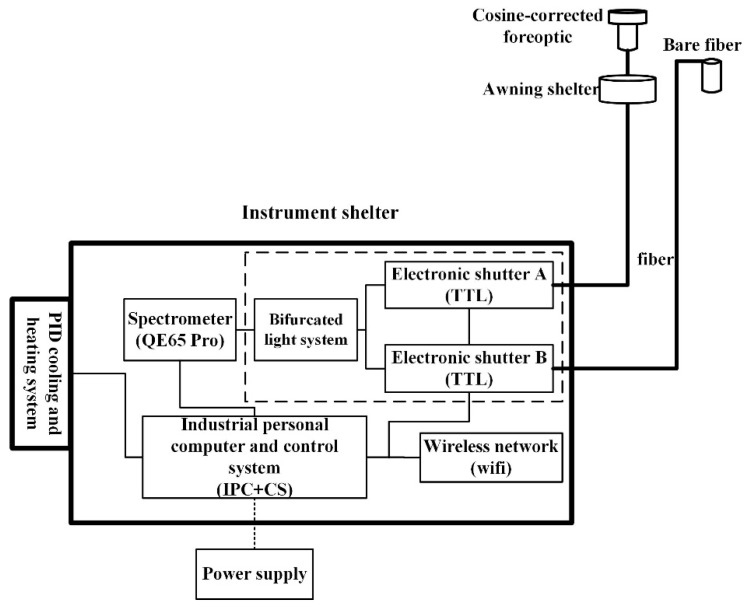
Schematic diagram of the configuration of the solar-induced chlorophyll fluorescence (SIF) measurement (SIFSpec) system. TTL and IPC indicate the electronic shutter and the industrial personal computer, respectively.

**Figure 2 sensors-19-03009-f002:**
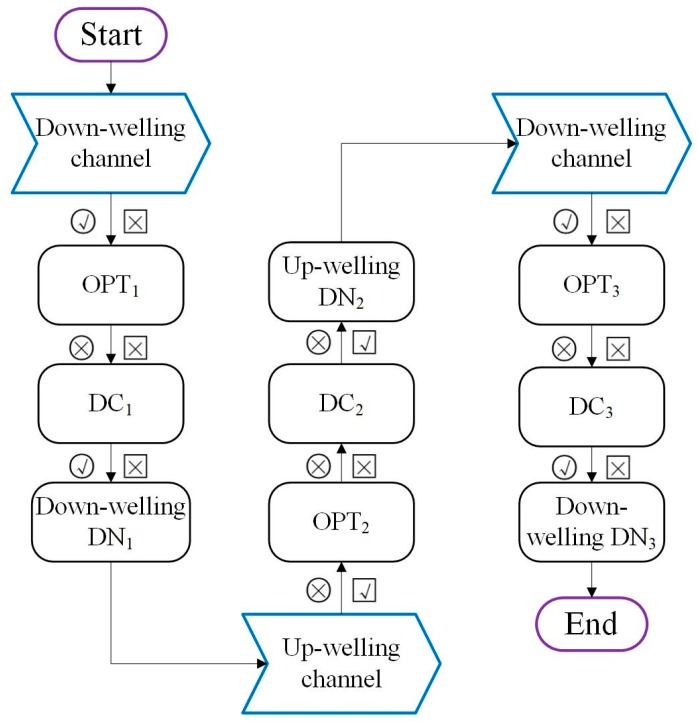
Flow diagram showing the data collection sequence used in the automated observation mode of the SIFSpec system. The circles and squares represent the downward and upward channels, respectively. The √ and × symbols indicate that the light channel is open and closed, respectively. ORT and DN indicate the optimization process of the integration time and the digital number measured by the spectrometer, respectively.

**Figure 3 sensors-19-03009-f003:**
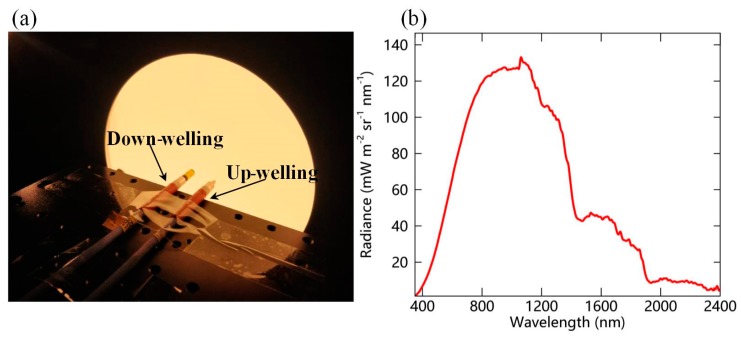
(**a**) Laboratory radiometric calibration of the SIFSpec system; (**b**) the radiance spectra of the light source used for calibrations.

**Figure 4 sensors-19-03009-f004:**
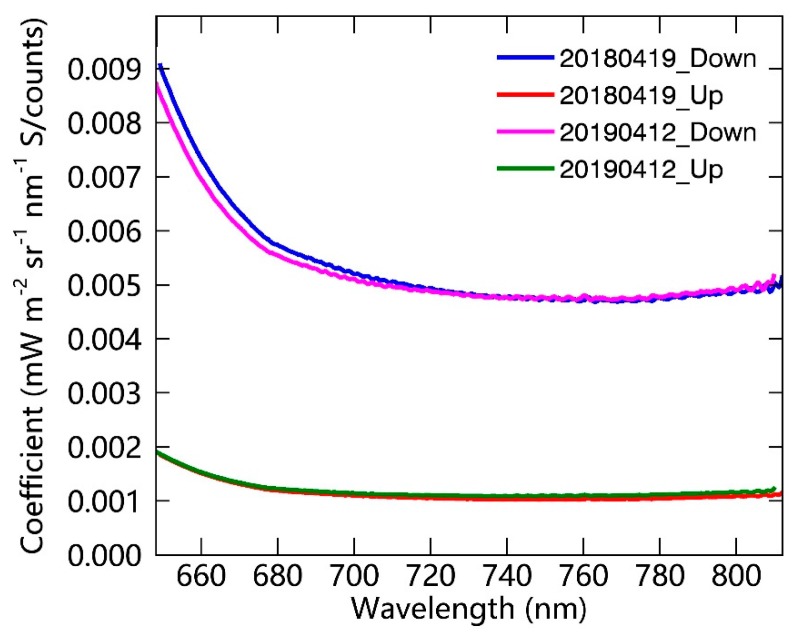
Variation with wavelength of the calibration coefficients of the upward and downward channels of the SIFSpec system at the DM site based on laboratory calibration.

**Figure 5 sensors-19-03009-f005:**
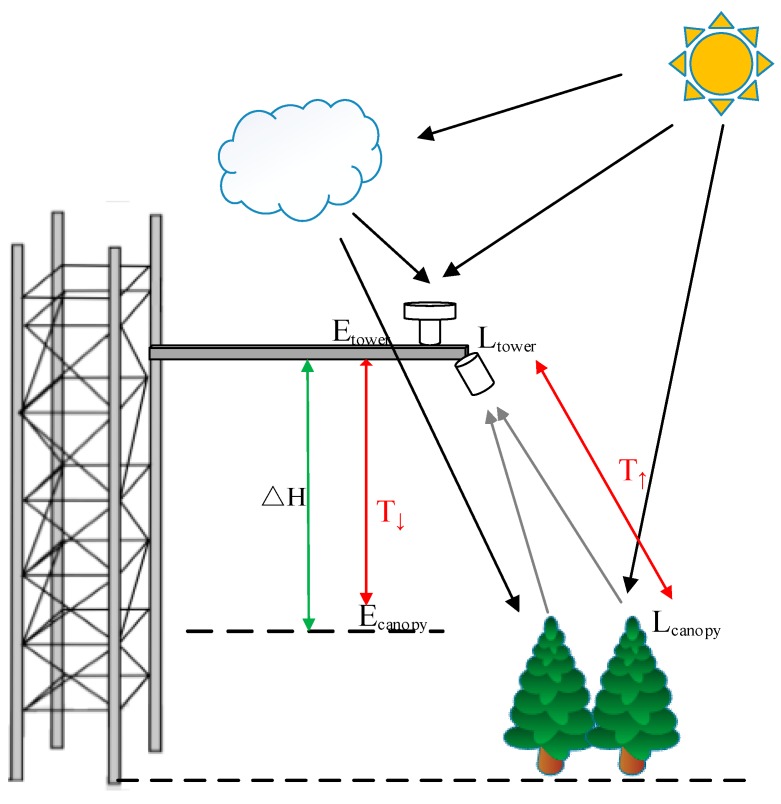
Schematic diagram of the atmospheric correction for tower-based SIFSpec system observations. $E_{tower}$ and $E_{canopy}$ are the downward irradiance at the sensor and top of the canopy; $L_{tower}$ and $L_{canopy}$ are the upward radiance arriving the sensor and at the top of the canopy; ΔH is the height between the canopy and the sensor; $T_{↑}$ and $T_{↓}$ are the upward and downward atmosphere transmittances between the canopy and the sensor, respectively.

**Figure 6 sensors-19-03009-f006:**
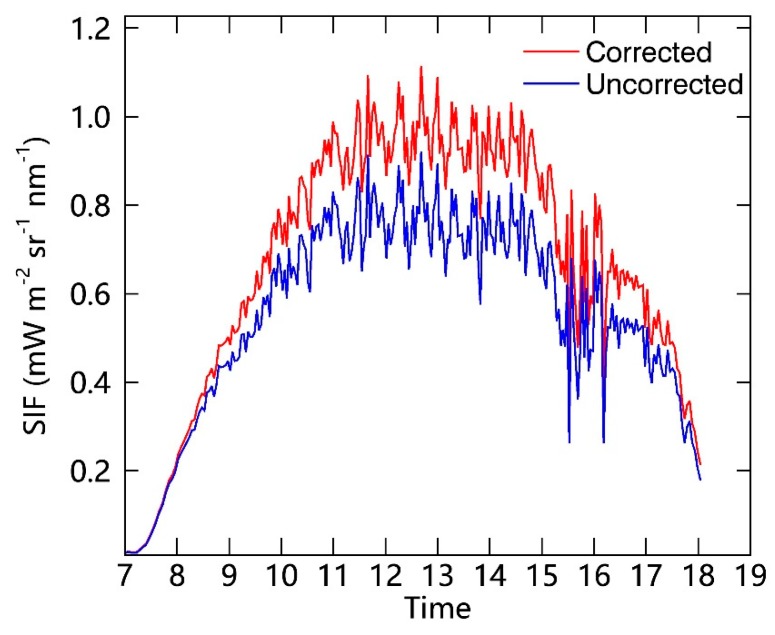
The diurnal variations in SIF results at the O_2_–A band before/after atmospheric correction as measured on 7 September 2017 at the DM site (corrected: red; uncorrected: blue).

**Figure 7 sensors-19-03009-f007:**
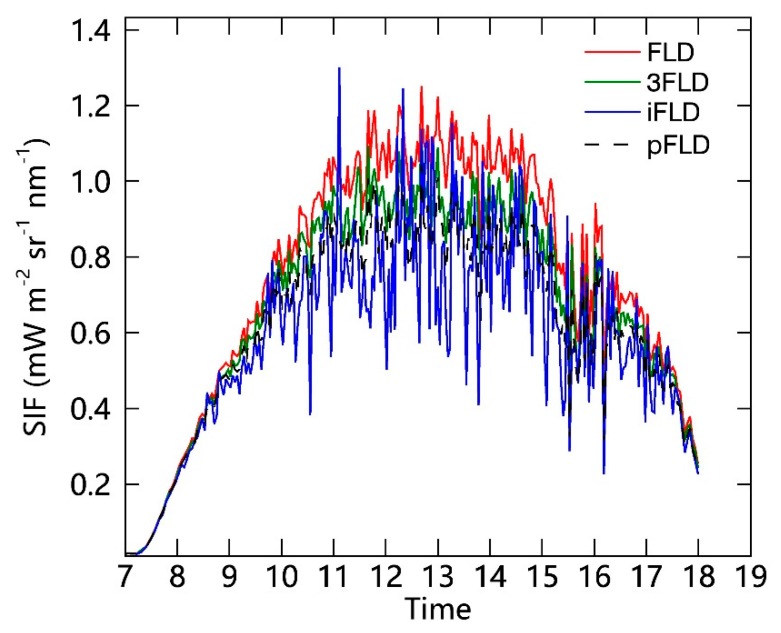
An example of the diurnal variations in SIF results at the O_2_–A band as measured on 7 September 2017 at the DM site using four different retrieval methods (Fraunhofer line discrimination (FLD): red; 3FLD: green; iFLD: blue; pFLD: black).

**Figure 8 sensors-19-03009-f008:**
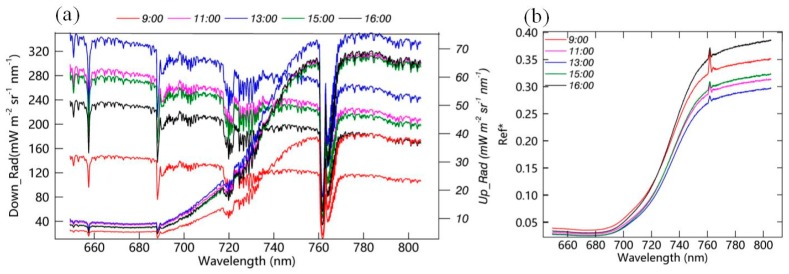
(**a**) Several examples of downward and upward radiances and (**b**) apparent reflectance for a clear-sky day (7 September 2017) at the DM site. The spectra with different colors represent the data measured at different times.

**Figure 9 sensors-19-03009-f009:**
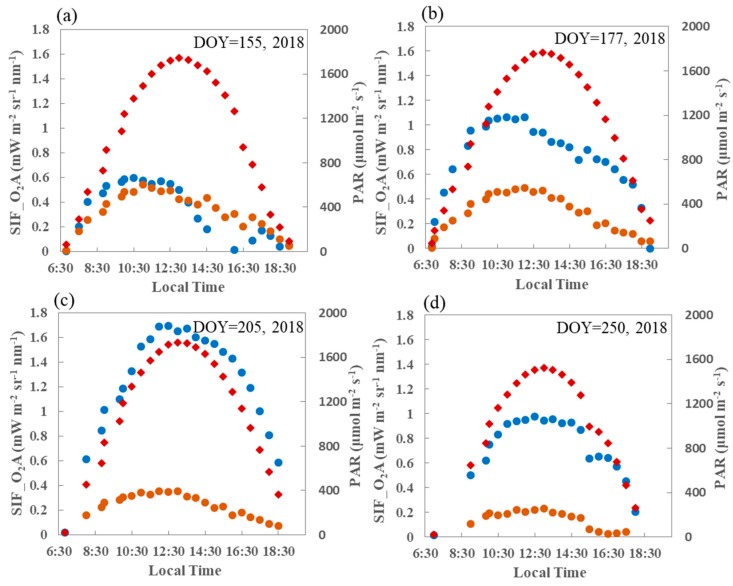
Diurnal variations in SIF at both the O_2_–A (blue) and O_2_–B (orange) bands and in photosynthetically active radiation (PAR) (red) at the DM site on four different days during the 2018 growing season. DOY—day of year.

**Figure 10 sensors-19-03009-f010:**
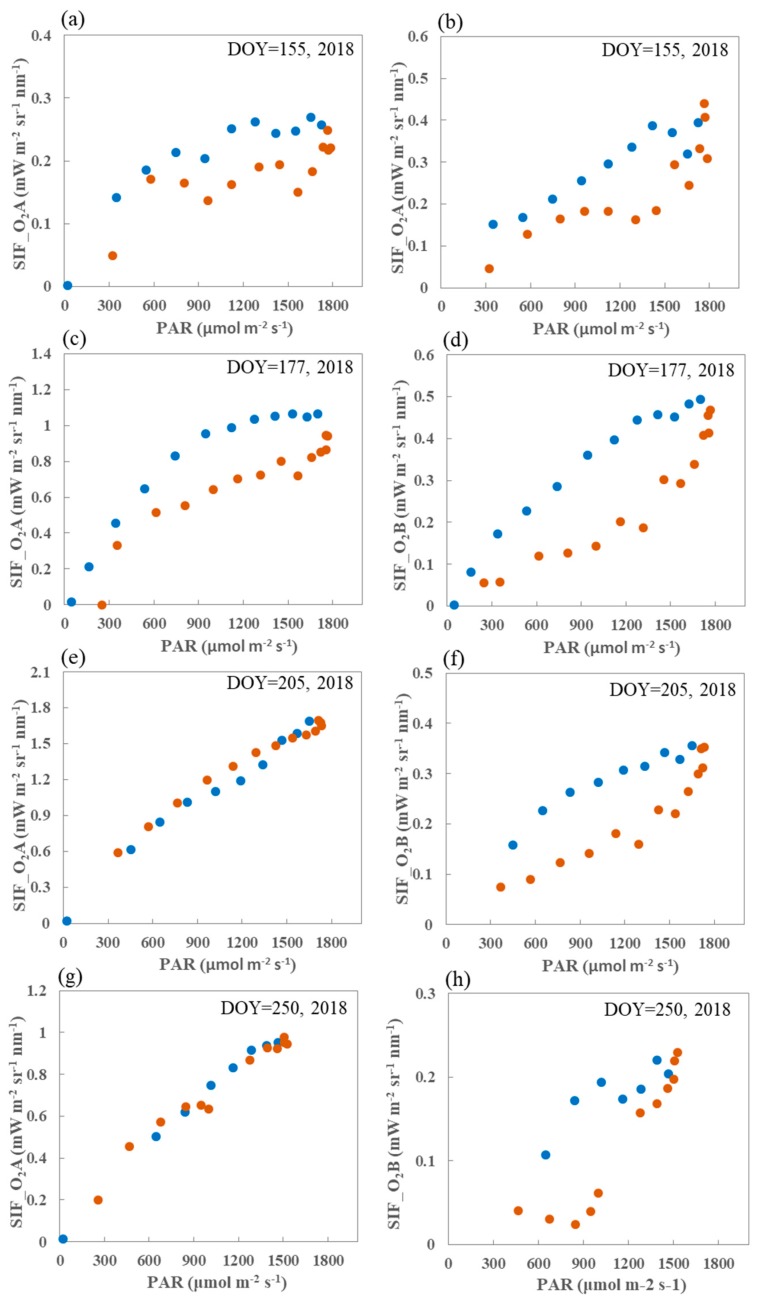
The diurnal response of SIF to PAR at both O_2_–A (left) and O_2_–B (right) bands at the DM site on four different days during the 2018 growing season. The blue and orange points indicate morning and afternoon, respectively.

**Figure 11 sensors-19-03009-f011:**
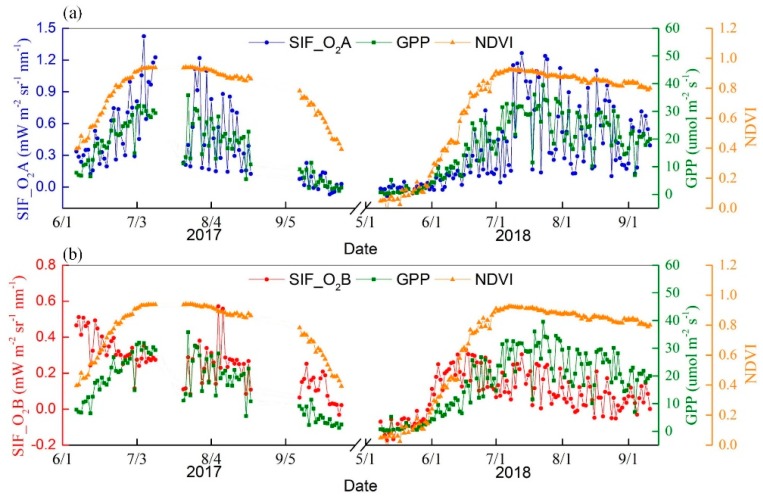
Seasonal variations in SIF at both the (**a**) O_2_–A (blue) and (**b**) O_2_–B (red) bands. Also shown are Normalized Difference Vegetation Index (NDVI) (orange) and flux gross primary production (GPP) (green) data obtained by the SIFSpec system at the DM site.

**Table 1 sensors-19-03009-t001:** Details of the eddy covariance (EC) flux sites with solar-induced chlorophyll fluorescence (SIF) measurement (SIFSpec) mounted on the tower.

Ecosystem Type	Site Name	ID	Latitude	Longitude	Height
Cropland	XiaoTangshan	XTS	40.1786 N	116.4432 E	4 m
	HuaiLai	HL	40.3489 E	115.7882 N	4 m
	DaMan	DM	38.8555 E	100.3722 N	25 m
	ShangQiu	SQ	34.5870 N	115.5753 E	12 m
	JuRong	JR	31.8068 N	119.2173 E	7 m
Forest	QianYanzhou	QYZ	26.7478 N	115.0581 E	32 m
	DingHushan	DHS	23.1733 N	112.5361 E	36 m
Grassland	XiLinhaote	XLHT	43.5513 N	116.6710 E	2.5 m
	HongYuan	HY	32.8404 N	102.5775 E	3 m
	ARou	AR	38.0444 N	100.4647 E	25 m

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
