# Peer review of "SIFSpec: Measuring Solar-Induced Chlorophyll Fluorescence Observations for Remote Sensing of Photosynthesis"

_sensors, 2019, doi:10.3390/s19133009_

Round 1
Reviewer 1 Report
I really enjoyed reading this paper, it is very well written, presented and highly topical as we navigate our way in SIF and explore its effectiveness. One of the key issues to date has been the instrument resolution used and sequence of processing. The novelty here is the authors used a well known system, apply the requisite corrections and then explore a range of SIF retrievals over multiple targets and over a time series. This is very thorough, and will be of much interest to the community. I don't have any major concerns over the paper but had a few suggestions:
The community doesn't fully realise what difference the atmospheric correction makes to the retrievals - so I would like to see a before and after graph showing the SIF before correction, and after. So perhaps have two graphs of Figure 7. Also the work of Neus Sabater (Remote Sensing) is not cited until very late on in the paper and should be added earlier perhaps as this was an in depth study into the impacts of atmosphere on the SIF. There is also the paper of Caroline J Nichol (again in Remote Sensing in 2019) which covers SIF and atmospheric correction. In fact the journal Remote Sensing has a special issue on SIF which has a number of papers which the authors would benefit from citing.
Minor points
The wording in Figure 1 is a bit grainy. Can this be sharpened?
In approx lines 72-79 please add citation of Nichol et al 2019 (and check those SIF papers published in 2019 in the Remote Sensing special issue)
Please remove author names before a numerical citation (for example on page 4, and 5)
Line 264 How regularly is the calibration conducted? The text says on "regular calibration". Please add this detail.
The axes in the Figures are very small. Please make a bit bigger
Author Response
Response to Comments from Reviewer 1
Reviewer: 1
I really enjoyed reading this paper, it is very well written, presented and highly topical as we navigate our way in SIF and explore its effectiveness. One of the key issues to date has been the instrument resolution used and sequence of processing. The novelty here is the authors used a well known system, apply the requisite corrections and then explore a range of SIF retrievals over multiple targets and over a time series. This is very thorough, and will be of much interest to the community.
Many thanks for the encouraging words.
I don't have any major concerns over the paper but had a few suggestions:
The community doesn't fully realise what difference the atmospheric correction makes to the retrievals - so I would like to see a before and after graph showing the SIF before correction, and after. So perhaps have two graphs of Figure 7.
Thank you for this comment. According to your suggestion, the SIF retrievals at O2-A band before/after atmospheric correction has been added in the Section 3.3.
‘And the tower-based SIF retrievals at O2-A band before/after atmospheric correction are presented in Figure 6. It shows that the SIF results are obviously higher after atmospheric correction than the uncorrected SIF results, especially for noon time.’
Figure 6. The diurnal variations in SIF results at O2-A band before/after atmospheric correction as measured on 7 September 2017 at the DM site (Corrected: red; Uncorrected: blue).
Also the work of Neus Sabater (Remote Sensing) is not cited until very late on in the paper and should be added earlier perhaps as this was an in depth study into the impacts of atmosphere on the SIF. There is also the paper of Caroline J Nichol (again in Remote Sensing in 2019) which covers SIF and atmospheric correction. In fact the journal Remote Sensing has a special issue on SIF which has a number of papers which the authors would benefit from citing.
Many thanks for this suggestion. According to your suggestion, the related papers and descriptions have been added in the Section 1 and References.
‘It is worth mentioning that the recent work has also addressed the importance of oxygen compensation on proximal sensing SIF retrieved at O2 band using Fraunhofer Line Discrimination (FLD) and Spectral Fitting Methods (SFM) [30]. And this method has been used in the field measurements for a boreal forest ecosystem at a height of about 18m to account for the energy reabsorption effects on SIF retrievals at O2-A band [31].’
‘31. Nichol, C.; Drolet, G.; Porcar-Castell, A.; Wade, T.; Sabater, N.; Middleton, E.; MacLellan, C.; Levula, J.; Mammarella, I.; Vesala, T.; Atherton, J., Diurnal and Seasonal Solar Induced Chlorophyll Fluorescence and Photosynthesis in a Boreal Scots Pine Canopy. Remote Sensing 2019, 11, (3).’
Minor points
The wording in Figure 1 is a bit grainy. Can this be sharpened?
Great thanks for this comment. According to your comment, Figure 1 has been revised and replaced with the revised figure in the manuscript.
Figure 1. Schematic diagram of the configuration of the SIFSpec system.
In approx lines 72-79 please add citation of Nichol et al 2019 (and check those SIF papers published in 2019 in the Remote Sensing special issue)
Great thanks for this suggestion. According to your suggestion, the related papers and descriptions have been added in the Section 1 and References.
‘It is worth mentioning that the recent work has also addressed the importance of oxygen compensation on proximal sensing SIF retrieved at O2 band using Fraunhofer Line Discrimination (FLD) and Spectral Fitting Methods (SFM) [30]. And this method has been used in the field measurements for a boreal forest ecosystem at a height of about 18m to account for the energy reabsorption effects on SIF retrievals at O2-A band [31].’
‘31. Nichol, C.; Drolet, G.; Porcar-Castell, A.; Wade, T.; Sabater, N.; Middleton, E.; MacLellan, C.; Levula, J.; Mammarella, I.; Vesala, T.; Atherton, J., Diurnal and Seasonal Solar Induced Chlorophyll Fluorescence and Photosynthesis in a Boreal Scots Pine Canopy. Remote Sensing 2019, 11, (3).’
Please remove author names before a numerical citation (for example on page 4, and 5)
Great thanks for this comment. According to your comment, this problem has been checked and revised throughout the manuscript.
Line 264 How regularly is the calibration conducted? The text says on "regular calibration". Please add this detail.
Thanks for this comment. The detail about the calibration frequency has been added in the Section 3.2.
‘In general, the calibration is conducted before the instrument installed for every growth period for cropland site and once a year for forest site.’
The axes in the Figures are very small. Please make a bit bigger
Thanks for this comment. All figures have been revised according to your comment. Considering the neatness of the manuscript, all the original figures are directly replaced with the revised figures in the manuscript.

Reviewer 2 Report
The manuscript is based on original work and contains all requested items. The advantageous of these work is that the experimental work was done at different far separated locations and on various species.
However, the work should consider a comparison between the developed system and others to show the advantages and disadvantages clearly.
Some minor typos are there and should be removed e.g. 2017 X2 in:
p.p1 {margin: 0.0px 0.0px 0.0px 0.0px; font: 9.0px Times} span.s1 {font: 10.0px Times}
Zhang, Y.; Wang, S.; Liu, L.; Ju, W.; Zhu, X. In ChinaSpec: a network of SIF observations to bridge flux 648 measurements and remote sensing data, Agu Fall Meeting, 2017; 2017.
Author Response
Response to Comments from Reviewer 2
Reviewer: 2
The manuscript is based on original work and contains all requested items. The advantageous of these work is that the experimental work was done at different far separated locations and on various species.
Many thanks for the encouraging words.
However, the work should consider a comparison between the developed system and others to show the advantages and disadvantages clearly.
Thank you for this comment. According to your comment, we have added another part about the selection of the spectral range for the automated system and discussed the advantages of the spectral range in the SIF observation system in the Section 5.4.
5.4 The selection of the spectral range for the SIFSpec system
At the present stage, although the spectral range of spectrometers used in the SIFSpec system is about from 650 nm to 800 nm, the other existing automated systems mainly cover the near-infrared (NIR) region (~730-780nm), as introduced in the above. Thus, the information of red SIF retrieval is not available for other SIF observation systems only covering NIR region. In practice, it is more difficult for red SIF signals to escape from canopy due to the strong re-absorption and scattering effects inside the vegetation canopy [61]. Therefore, the red SIF is much weaker and is also more difficult to be measured at canopy level than that at the NIR band, which may be the main reason to neglect the red band in the systems. However, the red SIF may be more directly linked to photosynthetic activity because it contains more information of the photosystem II [61-63]. In addition, although the spectral resolution will decrease for instrument with wider spectral range (concluding red and NIR bands) compared to shorter range (only concluding NIR band), which decreases from 0.15 nm to 0.34 nm, the integration time will be reduced and the SNR will increase to some extent due to the increased photons that reach each detector per unit time [27]. And the previous work has demonstrated that the difference of the retrieved SIF with spectral resolution of 0.1 nm and 0.3 nm at O2-A band can be negligible if tested using the 3FLD SIF retrieval method [49]. Furthermore, the red SIF signal will be much enhanced if it is downscaled from the canopy level to the photosystem level [64], which will may improve the correlation with photosynthetic productivity. Therefore, in this context, the red band is suggested to be concluded in the SIFSpec system.’
‘61. Porcar-Castell, A.; Tyystjärvi, E.; Atherton, J.; van der Tol, C.; Flexas, J.; Pfündel, E. E.; Moreno, J.; Frankenberg, C.; Berry, J. A., Linking chlorophyll a fluorescence to photosynthesis for remote sensing applications: mechanisms and challenges. Journal of Experimental Botany 2014, eru191.
62. Agati, G.; Cerovic, Z. G.; Moya, I., The Effect of Decreasing Temperature up to Chilling Values on the in vivo F685/F735 Chlorophyll Fluorescence Ratio in Phaseolus vulgaris and Pisum sativum: The Role of the Photosystem I Contribution to the 735 nm Fluorescence Band¶. Photochemistry and Photobiology 2000, 72, (1), 75-84.
63. Joiner, J.; Yoshida, Y.; Guanter, L.; Middleton, E. M., New methods for retrieval of chlorophyll red fluorescence from hyper-spectral satellite instruments: simulations and application to GOME-2 and SCIAMACHY. Atmospheric Measurement Techniques Discussions 2016, 1-41.
64. Liu, X.; Guanter, L.; Liu, L.; Damm, A.; Malenovský, Z.; Rascher, U.; Peng, D.; Du, S.; Gastellu-Etchegorry, J.-P., Downscaling of solar-induced chlorophyll fluorescence from canopy level to photosystem level using a random forest model. Remote Sensing of Environment 2018.’
Some minor typos are there and should be removed e.g. 2017 X2 in:
p.p1 {margin: 0.0px 0.0px 0.0px 0.0px; font: 9.0px Times} span.s1 {font: 10.0px Times}
Zhang, Y.; Wang, S.; Liu, L.; Ju, W.; Zhu, X. In ChinaSpec: a network of SIF observations to bridge flux 648 measurements and remote sensing data, Agu Fall Meeting, 2017; 2017.
Great thanks for this comment. Sorry for these technical problems. This problem has been checked and revised according to your comment.
‘32. Zhang, Y.; Wang, S.; Liu, L.; Ju, W.; Zhu, X. In ChinaSpec: a network of SIF observations to bridge flux measurements and remote sensing data, AGU Fall Meeting, 2017.’
Reviewer 3 Report
The sensor and network are well-designed and the authors do a very nice job of introducing the relevance and importance of the sensor, describing methods, and presenting their results. There were some updates suggested in these areas, which should be addressed for publication.
The majority of updates to writing need to be concentrated in the discussion. The discussion and summary section did not clearly or completely summarize the very positive results the authors presented. In the discussion and summary, the authors have some information about topics that potentially could be addressed in future work (provided they are things that could not be addressed in this study). It was difficult to understand the purpose of the points being made in the discussion, thus it was difficult to make specific suggested changes. It could be improved by briefly telling the reader what the specific results mean and then what they mean in the context of using the sensor and measurements going forward, possibly within individual sites or across the tower network, or in context of other studies. Additionally, there are really interesting paths for future research and future publications that could be suggested from the study and should be discussed (briefly) and this may be the direction the discussion is supposed to take the reader. If so, a few details added and explicitly stating this would rectify the problem. A few updates to the end of the paper with this in mind would help the reader take important information away.
The text (axis titles and labels) in the figures needs to be larger, as do the figures themselves. This would make deciphering much easier. Specific notes are on certain figures, but almost every figure needs this update and most of them need panels labelled (a,b,c,d,e...). Almost all figures need more description in the caption AND within the text of the manuscript. Although labelling the panels and referring to them may clear up the confusion.
Overall, the paper is very nice and is exciting and important research into SIF. I look forward to the next draft and publication.
Author Response
Response to Comments from Reviewer 3
Reviewer: 3
The sensor and network are well-designed and the authors do a very nice job of introducing the relevance and importance of the sensor, describing methods, and presenting their results.
There were some updates suggested in these areas, which should be addressed for publication.
Many thanks for the encouraging and critical words.
The majority of updates to writing need to be concentrated in the discussion. The discussion and summary section did not clearly or completely summarize the very positive results the authors presented. In the discussion and summary, the authors have some information about topics that potentially could be addressed in future work (provided they are things that could not be addressed in this study). It was difficult to understand the purpose of the points being made in the discussion, thus it was difficult to make specific suggested changes. It could be improved by briefly telling the reader what the specific results mean and then what they mean in the context of using the sensor and measurements going forward, possibly within individual sites or across the tower network, or in context of other studies.
Additionally, there are really interesting paths for future research and future publications that could be suggested from the study and should be discussed (briefly) and this may be the direction the discussion is supposed to take the reader. If so, a few details added and explicitly stating this would rectify the problem. A few updates to the end of the paper with this in mind would help the reader take important information away.
Many thanks for this constructive comment. According to your comment, both the conclusion and discussion sections were revised. We have added another part about the selection of the spectral range for the automated system and discussed the advantages of the spectral range in the SIFSpec in the Section 5.4.
‘6. Conclusion
Automated and long-term in-situ SIF measurements coordinated with meteorological and EC observations have a significant role in terrestrial ecosystem research. In particular, long-term in-situ SIF observations provide unprecedented opportunities for the direct validation of satellite-based and airborne-based SIF retrievals and the possibility of investigating the physiological link between SIF and GPP. In this paper, the automated and long-term in-situ SIF measurement system - SIFSpec used in the ChinaSpec network - was presented. Compared to most currently existing SIF observation system only covering NIR region, which can only retrieve NIR SIF signal, our SIFSpec can record both red and NIR SIF signals. We have demonstrated the reliability and practicability of the SIF measurements collected by the SIFSpec system using the observations made at the DM site over two growing cycles. The consistency between the diurnal variations in SIF and PAR at both the O2-A and O2-B bands throughout the entire canopy growth stage is essential to the psychological mechanism of SIF emitted by vegetation. The consistency between the seasonal variations in SIF and GPP demonstrates the important role of SIF in the exploration of the photosynthetic function of vegetation. ’
5.4 The selection of the spectral range for the SIFSpec system
At the present stage, although the spectral range of spectrometers used in the SIFSpec system is about from 650 nm to 800 nm, the other existing automated systems mainly cover the near-infrared (NIR) region (~730-780nm), as introduced in the above. Thus, the information of red SIF retrieval is not available for other SIF observation systems only covering NIR region. In practice, it is more difficult for red SIF signals to escape from canopy due to the strong re-absorption and scattering effects inside the vegetation canopy [61]. Therefore, the red SIF is much weaker and is also more difficult to be measured at canopy level than that at the NIR band, which may be the main reason to neglect the red band in the systems. However, the red SIF may be more directly linked to photosynthetic activity because it contains more information of the photosystem II [61-63]. In addition, although the spectral resolution will decrease for instrument with wider spectral range (concluding red and NIR bands) compared to shorter range (only concluding NIR band), which decreases from 0.15 nm to 0.34 nm, the integration time will be reduced and the SNR will increase to some extent due to the increased photons that reach each detector per unit time [27]. And the previous work has demonstrated that the difference of the retrieved SIF with spectral resolution of 0.1 nm and 0.3 nm at O2-A band can be negligible if tested using the 3FLD SIF retrieval method [49]. Furthermore, the red SIF signal will be much enhanced if it is downscaled from the canopy level to the photosystem level [64], which will may improve the correlation with photosynthetic productivity. Therefore, in this context, the red band is suggested to be concluded in the SIFSpec system.’
‘61. Porcar-Castell, A.; Tyystjärvi, E.; Atherton, J.; van der Tol, C.; Flexas, J.; Pfündel, E. E.; Moreno, J.; Frankenberg, C.; Berry, J. A., Linking chlorophyll a fluorescence to photosynthesis for remote sensing applications: mechanisms and challenges. Journal of Experimental Botany 2014, eru191.
62. Agati, G.; Cerovic, Z. G.; Moya, I., The Effect of Decreasing Temperature up to Chilling Values on the in vivo F685/F735 Chlorophyll Fluorescence Ratio in Phaseolus vulgaris and Pisum sativum: The Role of the Photosystem I Contribution to the 735 nm Fluorescence Band¶. Photochemistry and Photobiology 2000, 72, (1), 75-84.
63. Joiner, J.; Yoshida, Y.; Guanter, L.; Middleton, E. M., New methods for retrieval of chlorophyll red fluorescence from hyper-spectral satellite instruments: simulations and application to GOME-2 and SCIAMACHY. Atmospheric Measurement Techniques Discussions 2016, 1-41.
64. Liu, X.; Guanter, L.; Liu, L.; Damm, A.; Malenovský, Z.; Rascher, U.; Peng, D.; Du, S.; Gastellu-Etchegorry, J.-P., Downscaling of solar-induced chlorophyll fluorescence from canopy level to photosystem level using a random forest model. Remote Sensing of Environment 2018.’
The text (axis titles and labels) in the figures needs to be larger, as do the figures themselves. This would make deciphering much easier. Specific notes are on certain figures, but almost every figure needs this update and most of them need panels labelled (a,b,c,d,e...). Almost all figures need more description in the caption AND within the text of the manuscript. Although labelling the panels and referring to them may clear up the confusion.
Thank you for this comment. According to your comment, all the figures and the corresponding text and labels have been updated in the manuscript. And we added annotation in Figure 5 to make the figure clearly. Considering the neatness of the manuscript, all the original figures are directly replaced with the revised figures in the manuscript.
Figure 5. Schematic diagram of the atmospheric correction for tower-based SIFSpec system observations. and are the downward irradiance at the sensor and top of the canopy; and are the upward radiance arriving the sensor and at the top of the canopy; is the height between the canopy and the sensor; and are the upward and downward atmosphere transmittances between the canopy and the sensor, respectively.
Overall, the paper is very nice and is exciting and important research into SIF. I look forward to the next draft and publication
Many thanks for the encouraging words.
